

# On the magnitude and sensitivity of the QBO response to a tropical volcanic eruption

Flossie Brown[1], Lauren Marshall[2,*], Peter Haynes[3], Rolando Garcia[4], Thomas Birner[5,6], Anja Schmidt[2,5,6]

[1] College of Engineering, Mathematics and Physical Sciences, University of Exeter, Exeter, UK
[2] Yusuf Hamied Department of Chemistry, University of Cambridge, Cambridge, United Kingdom
[3] Department of Applied Mathematics and Theoretical Physics, University of Cambridge, Cambridge, UK
[4] National Center for Atmospheric Research, Boulder, USA
[5] Faculty of Physics, Meteorological Institute, Ludwig-Maximilians-University Munich, Munich, Germany
[6] Institute of Atmospheric Physics (IPA), German Aerospace Center (DLR), Oberpfaffenhofen, Germany
*now at: Department of Earth Sciences, Durham University, Durham, UK

Correspondence to: Flossie Brown (fb428@exeter.ac.uk)

**Abstract**

Volcanic eruptions that inject sulphur dioxide into the stratosphere have the potential to alter large-scale circulation patterns, such as the quasi-biennial oscillation (QBO), which can affect weather and transport of chemical species. Here, we conduct simulations of tropical volcanic eruptions using the UM-UKCA aerosol-climate model with an explicit representation of the QBO. Eruptions emitting 60 Tg of $SO_2$ (i.e., 1815 Mt. Tambora-magnitude) and 15 Tg of $SO_2$ (i.e., 1991 Mt. Pinatubo-

magnitude) were simulated at the equator initiated during two different QBO states. We show that tropical eruptions delay the progression of the QBO phases, with the magnitude of the delay dependent on the initial wind shear in the lower stratosphere and a much longer delay when the shear is easterly than when it is westerly. The QBO response in our model is driven by vertical advection of momentum by the stronger tropical upwelling caused by heating due to the increased volcanic sulfate aerosol loading. Direct aerosol-induced warming with subsequent thermal wind adjustment, as proposed by previous studies,

is found to only play a secondary role. This interpretation of the response is supported by comparison with a simple dynamical model. The dependence of the magnitude of the response on the initial QBO state results from differences in the QBO secondary circulation. In the easterly shear zone of the QBO, the vertical component of the secondary circulation is upward and reinforces the anomalous upwelling driven by volcanic aerosol heating, whereas in the westerly shear zone the vertical component is downward and opposes the aerosol-induced upwelling. We also find a change to the latitudinal structure of the

QBO, with the westerly phase of the QBO strengthening in the hemisphere with the lowest sulfate aerosol burden. Overall, our study suggests that tropical eruptions of Pinatubo-magnitude or larger could force changes to the progression of the QBO, with particularly disruptive outcomes for the QBO if the eruption occurs during the easterly QBO shear.





## 1. Introduction

When explosive volcanic eruptions inject sulfur dioxide ($SO_2$) into the stratosphere, the effects can be global. In the last 50 years, eruptions of El Chichón (1982) and Mt. Pinatubo (1991) have released large amounts of $SO_2$ into the stratosphere, altering temperature distributions and circulation patterns in both the stratosphere and the troposphere (Labitzke & McCormick, 1992; Robock, 2000; Stenchikov et al., 2002). An understanding of the effects of such explosive eruptions is key for accurate weather and climate predictions. Furthermore, the effects of Pinatubo-magnitude eruptions are sufficiently large relative to natural atmospheric variability that they provide very useful tests of modelling capability, particularly with regard to the stratosphere.

In this paper we focus on the possible effects of $SO_2$ injections from large-magnitude tropical eruptions on the stratospheric Quasi-biennial Oscillation (QBO), which is an oscillation of the tropical zonal winds, between easterly and westerly, in the altitude range 16–40 km and with a period of approximately 28 months (Baldwin et al., 2001). The easterly and westerly phases propagate downwards towards the tropopause. The pattern of descending, alternating phases is driven by atmospheric waves, which provide easterly and westerly zonal-mean forces (Lindzen and Holton, 1968) and the speed of descent is determined by a balance between the wave-driven acceleration and upward advection of zonal-mean momentum by the Brewer-Dobson circulation (BDC) (Baldwin et al., 2001). The variation in tropical winds associated with the QBO influences subtropical and extratropical wave activity and the polar vortex (e.g., Holton and Tan, 1980; Anstey and Shepherd, 2014). Changes to the extratropical stratospheric circulation can in turn lead to changes near the surface, so the QBO is important for seasonal weather forecasting (Scaife et al., 2014). The QBO may also affect weather systems in the tropics, e.g., by modulating the Madden-Julian Oscillation (Son et al., 2017).

Following a large, explosive volcanic eruption, $SO_2$ injected into the stratosphere is oxidised to form sulfuric acid aerosol particles (referred to as volcanic sulfate aerosol throughout). The increase in volcanic sulfate aerosol burden causes radiative heating of the lower stratosphere by absorption of infra-red (IR) from the surface and near-IR from solar radiation (Robock, 2000). After the eruption of Mt. Pinatubo in the Philippines (15.13° N, 120.35° E) in 1991, tropical stratosphere temperatures were observed to increase by up to 3.5 K above the mean (Labitzke & McCormick, 1992) and the upwards BDC strengthened in the tropics (Kinne et al., 1992). Climate model simulations of the Mt. Pinatubo eruption suggest that the heating due to the aerosol was about 0.2 K day$^{-1}$ at 30 hPa in the tropics (Rieger et al., 2020).

Labitzke (1994) noted that the westerly phase (measured by winds in the lower stratosphere) of the QBO after the Mt. Pinatubo eruption in 1991 was remarkably long, but did not discuss whether this was caused by the eruption. Two other large-magnitude eruptions (El Chichón in 1982 and Mt. Agung in 1963) have occurred since QBO measurements began in 1953 (Naujokat and Naujokat, 1986) and indeed Dunkerton (1983) identified a possible QBO perturbation caused by the Mt. Agung eruption. However the significant cycle-to-cycle variability in the QBO signal and the small number of relevant eruptions means that



observations alone are not sufficient to conclude an eruption effect. Furthermore, in the past three decades there have been
only a small number of model studies (Brenna et al., 2021, DallaSanta et al., 2021) attempting to obtain a more general
understanding of the effect of volcanic eruptions on the QBO.

There have been several recent studies investigating the response of the QBO to geoengineering through artificial injection
of $SO_2$ or sulfate into the stratosphere. Sulfate geoengineering is, in effect, equivalent to a sustained volcanic eruption,
although somewhat different to a short-lived, explosive eruption. Such studies (e.g., Aquila et al., 2014; Niemeier and
Schmidt, 2017; Richter et al., 2017; Tilmes et al., 2018; Franke et al., 2021) typically find that continuous $SO_2$ injection at
the equator over a period of several years causes the overall period of the QBO to increase, typically through increased
duration of westerlies in the lower stratosphere, which in some simulations becomes permanent so that the QBO disappears.
These studies (see in particular Aquila et al. 2014 and Franke et al. 2021) identify two distinct mechanisms as, in
combination, responsible for the QBO response to sulfate geoengineering. The two mechanisms may be termed (i) the
'thermal wind balance mechanism' and (ii) the 'upwelling mechanism'. In the thermal wind balance mechanism, changes in
temperature caused by the aerosol heating must, to satisfy thermal wind balance, be accompanied by changes in winds,
manifested by changes in the QBO. In the upwelling mechanism, aerosol heating drives a change in vertical velocity in the
tropical lower stratosphere which, through its effect on vertical advection of momentum, leads to a change in the progression
of the QBO. The naming of (i) and (ii) is chosen to reflect terminology of previous authors but it is important to note that
thermal wind balance of the zonal-mean temperature and zonal wind fields is always approximately realized, even in the
vicinity of the equator. The fundamental difference between mechanisms (i) and (ii), then, is how volcanic heating is viewed
as affecting the QBO: Mechanism (i) emphasizes direct modification of the temperature field by aerosol heating, whereas (ii)
acknowledges that temperature perturbations are not efficiently forced by heating near the equator and highlights the impact
of vertical velocity perturbations on the momentum budget of the QBO.

As sulfate geoengineering studies involve continuous injection over several years, the forcing is maintained and a new
statistical equilibrium is reached in which the QBO characteristics such as period and amplitude are different from those in the
unperturbed atmosphere. In contrast to geoengineering scenarios, the forcing from explosive volcanic eruptions typically
decays over 1–3 years. DallaSanta et al. (2021) have recently described multi-model simulations of Pinatubo-magnitude
eruptions. They considered the effect of the eruption on the period and amplitude of the following QBO cycle and found it to
be dependent on the QBO state at the time of eruption, characterising the post-eruption state as favouring lower stratospheric
westerlies and hence prolonging the period if this was the initial state and reducing it if the initial state had lower stratospheric
easterlies. Across the different models they considered, DallaSanta et al. (2021) identified a coherent dynamical response to
the aerosol injection and concluded that the effect on the QBO was primarily mediated by the upwelling mechanism. Another
recent study by Brenna et al. (2021) considered the effect of tropical super-eruptions (those having total $SO_2$ emissions of 1000
Tg, i.e., about 60 times those of Pinatubo). The response, an easterly state lasting for five years followed by a periodic QBO





with a longer period than before, was found to be independent of the initial QBO state and was attributed to the thermal wind balance mechanism, although the upwelling mechanism is also referred to. The formation of easterlies in the initial stages of the disruption caused by the super-eruption is distinctly different to the QBO response to the Pinatubo-magnitude simulations by DallaSanta et al. (2021) and to sulfate geoengineering simulations.

Any disruption of the QBO by volcanic eruptions is arguably best considered as a temporary disruption analogous to those observed in 2015 and 2020 (Osprey et al., 2016, Anstey et al., 2021) and analysed by considering the detailed time evolution of the equatorial zonal winds rather than by, e.g., a change in QBO period. During those observed disruptions, the descent of the westerly phase in the lower/mid stratosphere was interrupted by the appearance of easterlies which then descended, followed by the westerlies, as the QBO resumed its normal character. These disruptions are believed to be due to anomalously
strong propagation of large-scale waves from the extratropics into the tropics, and therefore very different in their mechanism from any effect of volcanic eruptions, but they are evidence that short-term modulation of the QBO momentum budget can cause observable changes in the QBO progression. We also consider that the dispersion of the aerosol may be affected by the circulation at the time of the eruption (Niemeier and Schmidt, 2017). The time-evolving dispersion of the aerosol is likely to influence the forcing, and therefore the circulation patterns. This could lead to differences in the forcing and response between
eruptions initiated during different QBO states and seasons.

Here we use an aerosol model coupled to a general circulation model, described in Sect. 2.1, to investigate changes to the QBO in response to a forcing from tropical volcanic eruptions releasing 15 Tg and 60 Tg of $SO_2$. An eruption releasing 60 Tg of $SO_2$ is comparable to the 1815 Mt. Tambora eruption, with an expected recurrence frequency of one eruption every 100 to 1000 years. The 15 Tg case is comparable to the 1991 Mt. Pinatubo eruption, with an expected recurrence frequency of one
eruption every 30 to 40 years. Simulation details and definitions of the QBO states used in this study are described in Sect. 2.2 and Sect. 2.3, respectively. Interpretation of the aerosol model results is aided by a simplified dynmical model of the QBO, described in Sect. 2.4.

We explore the following questions: does the state of the QBO or season of eruption have an effect on the QBO response (Sect.
3.1)? To what extent can the QBO response be explained by either the 'thermal wind balance mechanism' or the 'upwelling mechanism' (Sect. 3.2)? Is there an effect on the QBO response from the dispersion of the aerosol during eruptions in different states and seasons (Sect. 3.3)? In Sect. 4, we argue that the primary effect of an eruption on the QBO can be determined by considering the 'upwelling mechanism' and the implication of the increased vertical velocity on the QBO momentum balance.





## 2. Methods

### 2.1 Description of the UM-UKCA model

We use the UM-UKCA interactive stratospheric aerosol model, which consists of the U.K. Met Office Unified Model (UM) general circulation model (HadGEM-GA7.1) coupled with version 11.2 of the U.K. Chemistry and Aerosol scheme (UKCA). The model is run with a year-2000 atmosphere-only time-slice setup, with prescribed sea surface temperatures and sea-ice fraction and depth. The resolution is 1.25° (latitude) by 1.875° (longitude) with 85 vertical levels up to 85 km. The model configuration is based on the UM-UKCA 11.2 release version but with explosive eruptions inacted and a correction applied that reduces the number of nucleation mode aerosol particles, previously erroneously high (Ranjithkumar et al. 2021; Mulcahy et al. 2022).

UKCA contains the Global Model of Aerosol Processes (GloMAP; Mann et al., 2010; 2012) modal aerosol scheme, with aerosol simulated in 7 log-normal modes, and whole-atmosphere chemistry (Archibald et al., 2020). Volcanic eruptions are simulated by emitting $SO_2$ in one grid box at a certain latitude and longitude; the $SO_2$ is oxidised to $H_2SO_4$ vapour and sulfate aerosol is subsequently formed via nucleation and condensation and evolves via coagulation, sedimentation and wet and dry deposition. The radiation scheme accounts for aerosol radiative heating. UKCA includes gravity wave drag with parameterized orographic (Webster et al., 2003) and non-orographic components (Scaife et al., 2002).

This model has an internally generated QBO with amplitudes of up to 20 m s$^{-1}$ for the westerlies and up to −40 m s$^{-1}$ for the easterlies (Morgenstern et al., 2009) that compare well to observations (Naujokat, 1986). However, the average period length is ~50 months (Morgenstern et al., 2009), which is almost twice as long as the observed QBO period of 28 months (Schenzinger et al., 2017).

### 2.2 Model simulations and reanalysis data used in this study

We investigate the response of the QBO in UM-UKCA for two different eruption magnitudes, different initial states for the QBO shear zone at 30 hPa, and different seasons at the time of the eruption. Eruptions are simulated during states with easterly shear (e-QBO) and westerly shear (w-QBO) at 30 hPa chosen from a 9-year control simulation; the e-QBO and w-QBO states are defined in Sect. 2.3. We specified injections of 15 Tg and 60 Tg of $SO_2$, representative of large (Mt. Pinatubo) and very large (Mt. Tambora) eruptions. We present here the results from eruptions in July; however, we also performed a sensitivity test in which we simulated the same eruptions in January to investigate the response of the QBO to the season of eruption. The results for the January simulations are shown in the Supplementary information since, in general, we identify the same trends in regardless of season (see Text S2: Results from eruptions in January). An exception is that changes to the latitudinal structure of the QBO occur in the opposite hemisphere for eruptions in January compared to July. The simulations in January have different initial conditions to those in July so, although we do not have ensemble members for the perturbed cases, we use the





similarity in response between different seasons to increase confidence in our results. The simulations were run for 36 months
and the volcanic $SO_2$ was released between 18–20 km altitude at 140° E longitude and 0° N latitude.

The response of the QBO to volcanic eruptions, as measured by the difference between two simulations with and without the
perturbation associated with the eruption included, needs to be distinguished from differences among unperturbed simulations
that arise from internal dynamical variability (i.e., the QBO might be affected by such variability in tropical or in extratropical
dynamics). To address this point, we conducted two small ensembles of three control simulations without volcanic emissions;
one ensemble started during an e-QBO and one started during a w-QBO. Within each ensemble, one simulation used the same
initial conditions as the simulations containing the eruptions. For the other two control simulations, we perturbed the initial
conditions very slightly (at the bit level), with the expectation that these perturbations would grow through the simulation and
therefore provide an indication of the magnitude of internal variability in the unforced simulation.

To clarify the mechanisms driving the response of the QBO to eruptions we use a 2-D dynamical model (described in Sect.
2.4 below). Firstly, we use the 2-D model to assess the contribution from the 'thermal wind balance mechanism' and the
'upwelling mechanism' (both described in Sect. 2.4). Secondly, we test whether the primary differences in QBO responses
during different QBO states can be explained by the dynamical properties of the QBO (described as part of the dynamical
framework in Sect. 2.3) or the different transport of the aerosol under those conditions. To separate these possibilities, we
compare the UM-UKCA model results, which includes feedbacks between the aerosol and circulation, to results from the 2-D
model that does not include these feedbacks. The heating rate prescribed for the 2-D model does not consider how differences
in QBO state may affect the circulation of the aerosol nor does it vary with season. It therefore enables us to quantify the
response to eruptions if there were no effect on the heating from differences in the aerosol distribution between QBO states
and seasons.

To investigate the latitudinal structure of the QBO and any changes to it, we aim to identify the centre of the QBO westerly
phase by selecting the maximum westerly zonal mean wind in the region 10 hPa to 70 hPa between latitudes 15° N and 15° S
for each month. The latitude of this maximum zonal mean zonal wind is compared over time between simulations. To identify
whether the latitude of the maximum QBO westerlies is well represented in our model, and whether the change in latitude after
an eruption is significant, we compare to monthly reanalysis data from NASA MERRA-2 for the period 1980 to 2020
(Bosilovich, 2015). The 95[th] percentile of the observed latitude range is used to determine the significance of any changes to
the QBO latitudinal structure that occur after a simulated eruption.

## 2.3. e-QBO and w-QBO definition and the dynamical framework

We investigate the response of the QBO depending on the QBO state at the time of eruption. Often, the QBO state is
characterized by the QBO phase or vertical shear at a certain pressure (usually 30 to 50 hPa) although the definition is not
always consistent among papers (Wei et al., 2021). Here, we define the QBO state by the vertical shear of the zonal mean





zonal wind at 30 hPa. We define an e-QBO as having an easterly shear at 30 hPa (and therefore easterlies above and westerlies below 30 hPa) and a w-QBO as having a westerly shear at 30 hPa (and therefore westerlies above and easterlies below 30 hPa) (Fig. 1). In this paper, we also use the terms westerly and easterly phase to refer to the descending wind system of the QBO. For example, in a simulation initialized with an (unperturbed) w-QBO *state*, the westerly *phase* would be expected to descend with time.

The likely effect on the QBO of the injection of volcanic aerosol in the tropical lower stratosphere can be described using the long-established dynamical theory for the QBO itself and for the stratospheric meridional circulation. A first key point is that, whilst the mechanism for the QBO, including an explanation for the downward propagation of the QBO wind anomalies, can be captured in a 1-D model that considers variation only in the vertical, important features of the observed QBO follow only if the latitudinal structure is included. In particular, considering that approximate zonal-mean thermal wind balance must hold,
it follows that the easterly shear zone must have a negative temperature anomaly and the westerly shear zone must have a positive temperature anomaly at the equator relative to the long-term mean temperature (Plumb and Bell, 1982, Huang et al., 2006; Punge et al., 2009). Furthermore, since the time scales of the QBO anomalies are much longer than radiative time scales, these temperature anomalies must be maintained against the effect of radiative heating or cooling by vertical motion, with tropical upwelling (and compensating subtropical downwelling) in the easterly shear zone and tropical downwelling (and
compensating subtropical upwelling) in the westerly shear zone (Baldwin et al., 2001; Minschwaner et al., 2016). These patterns of upwelling and downwelling associated with the QBO are superimposed on the long-term time mean BDC upwelling. Figure 1 shows QBO states with easterly and westerly shear zones (Fig. 1a, 1b) and the associated dynamical properties, following the classic picture of Plumb and Bell (1982). The upwelling and downwelling circulations have a significant effect on the QBO winds, through the corresponding vertical advection of momentum (see equation (2) below) and
lead to well-known asymmetry in the downward propagation of shear zones, with slower descent of easterly shear zones and more rapid descent of westerly shear zones. This asymmetry tends to be stronger in the lower stratosphere than the middle stratosphere.

**2.4. 2-D Dynamical model description**

To understand the effect of volcanic aerosol heating on the QBO it is necessary to consider both the zonal-mean thermodynamic
and zonal momentum equation, and the coupling between these equations through the zonal-mean continuity equation and the zonal-mean meridional momentum equation (which yields the thermal wind equation). This system of equations has been much studied (e.g., Plumb 1982, Garcia 1987, Haynes et al., 1991, Ming et al., 2016) and has the well-known property that an applied force or an applied heating leads to responses both in zonal velocity $u$ and in temperature $T$, with these responses depending on latitude and on processes such as frictional or radiative damping. What is particularly relevant here is that near
the equator most of the response to an imposed heating appears as a vertical velocity, not as a change in temperature. Furthermore, the associated circulation in the meridional plane is very ineffective in driving changes in the zonal velocity $u$



near the equator because the resulting Coriolis force is very small. Indeed, this is the reason why, in the early stages of the search for a dynamical mechanism for the QBO itself, the hypothesis that the wind anomalies might be explained as a response to heating anomalies had to be rejected (Wallace and Holton, 1968).

Correspondingly, a mechanism for modification of QBO winds by volcanic aerosol heating that is a consequence of temperature changes directly caused by that volcanic heating (what we refer to as the thermal wind balance mechanism) will be very ineffective at low latitudes. Instead, the primary response to the heating will be a change in upwelling and the effect on zonal winds will be felt through the vertical momentum advection term in the zonal-mean momentum equation. This principle was demonstrated by Dunkerton (1983) who presented a 1-D model of volcanic impact on the QBO in which the

volcanic impact was imposed simply by adding a specified anomaly in vertical velocity. However, in many of the recent studies of geoengineering or, indeed, volcanic effects the principle has been obscured by invoking a version of the 'thermal wind balance mechanism'. Our argument in this paper is that the primary effect of an eruption on the QBO can be assessed by asking: (A) what will be the resulting change in tropical upwelling close to the equator? (B) What will be the implication of the upwelling for the QBO momentum balance?

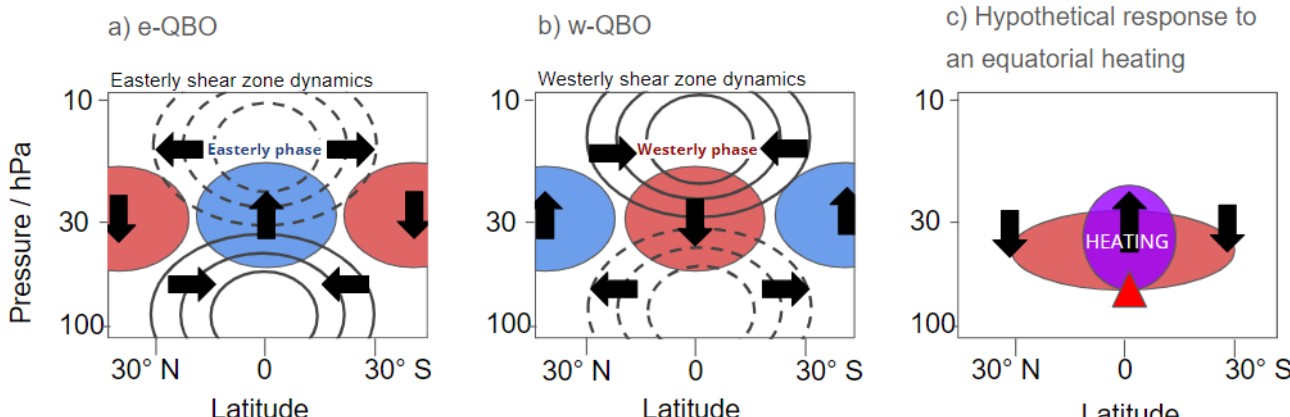


**Figure 1: Schematic temperature and circulation anomalies (a) associated with a lower stratospheric easterly shear zone and upper stratospheric westerly shear zone, (b) associated with a lower stratospheric westerly shear zone and upper stratospheric easterly shear zone, and (c) in response to heating from a volcanic SO₂ injection at the equator at the location indicated with a red triangle. Panels (a) and (b) are adapted from Plumb and Bell (1982). The black line contours in (a) and (b) represent positive and negative**
**zonal winds in solid and dashed lines, respectively. Positive temperature anomalies (filled shading) are shown in red and negative temperature anomalies are shown in blue. Arrows indicate circulation anomalies relative to the climatological average. The purple circle in panel (c) indicates the heating rate anomaly from the volcanic sulfate, which is narrow compared to the temperature anomaly.**

To illustrate and support these arguments, we use a simple model, based on the 2-D dynamical equations for a zonally
symmetric atmosphere, which incorporates a simple gravity wave parametrization and hence allows a QBO. The model equations are (e.g., Andrews et al 1987):





$$\frac{\partial T}{\partial t} + \frac{v}{a}\frac{\partial T}{\partial \varphi} + w\frac{\kappa\, T_b}{H} = Q_{volc} + Q_{rad}[T; z] \qquad (1)$$

$$\frac{\partial u}{\partial t} + v\left(\frac{1}{a\cos\varphi}\frac{\partial(u\cos\varphi)}{\partial\varphi} - 2\Omega\sin\varphi\right) + w\frac{\partial u}{\partial z} = G_{wave}[u; z] + G_{friction} \qquad (2)$$

$$2\Omega\,\sin\varphi\,u = -\frac{1}{a}\frac{\partial\Phi}{\partial\varphi} \qquad (3)$$

$$\frac{\partial\Phi}{\partial z} = \frac{R\,T}{H} \qquad (4)$$

$$\frac{1}{a\cos\varphi}\frac{\partial}{\partial\varphi}(v\cos\varphi) + \frac{1}{\rho_0}\frac{\partial}{\partial z}(\rho_0 w) = 0 \qquad (5)$$

where every variable is understood to be independent of longitude.

Notation is standard with $\varphi$ and $z$ being respectively latitude and log-pressure height (defined by $z = H\ln(p_0/p)$, where $p$ is pressure, $p_0$ is a reference pressure taken to be 1000 hPa, $H$ is a scale height, taken to be 7 km, and $\rho_0 = e^{-z/H}$). $R$ is the
gas constant and $a$ is the radius of the Earth. The model domain extends from pole to pole in latitude and from $z = 0$ to $z = 80$ km in height. The dynamical variables $u$, $v$, $w$, $T$ and $\Phi$ all represent anomalies with respect to a resting background state in which the temperature takes the constant value $T_b$. $u$ is the zonal velocity and $(v, w)$ the latitudinal and vertical components of the residual mean circulation (all measured in m s$^{-1}$). $T$ is the temperature anomaly and $\Phi$ is the corresponding geopotential anomaly. In the thermodynamic equation (1) $Q_{volc}$ is the volcanic aerosol heating with units of K
s$^{-1}$. $Q_{rad} = -\alpha T$ is a radiative relaxation that is proportional to the temperature anomaly $T$. The radiative relaxation time scale, $\alpha^{-1}$, is assumed to be 20 days throughout the whole model domain for simplicity. The term $w\,(\partial T/\partial z + \kappa T/H)$ (i.e., vertical advection of the potential temperature anomaly) is neglected. This follows Dunkerton (1991, see Section 2a) in his study of the height-latitude structure of the QBO in an idealised model, and thereby avoids numerical difficulties associated with the equations becoming ill-posed. Again following Dunkerton (1991), Eq. (3) represents geostrophic balance
rather than gradient wind balance, a justifiable approximation in the tropics. In the momentum equation (2) $G_{wave}[u; z]$ is a wave force that at each level depends on the vertical profile of $u$ and is specified according to the two-wave parametrization of Plumb (1977) as

$$G_{wave}[u, z] = -\frac{\partial}{\partial z}(F_1[u, z] + F_2[u, z]) + \frac{1}{\rho}\frac{\partial}{\partial z}v\rho\frac{\partial u}{\partial z}$$

where



$$F_i[u,z] = 0 (z < z_{launch})$$ (6)

$$F_i[u,z] = F_i^{(0)} \exp\{-\int_{z_{launch}}^{z} \frac{\alpha_i dz'}{(u-c_i)^2}(z > z_{launch})\}$$ (7)

and $\nu$ is a vertical diffusivity. The wave phase speeds, $c_i$ , momentum fluxes $F_i^{(0)}$ and the constants $\alpha_i$ controlling the decay rates were chosen simply to give a QBO-like oscillation similar to that observed, with $c_1 = -c_2 = 10$ m s$^{-1}$, $F_1^{(0)} = -F_2^{(0)} = 5 \times 10^{-4}$ m$^2$ s$^{-2}$ and $\alpha_1 = \alpha_2 = 10^{-2}$ m s$^{-2}$ . The launch height $z_{launch}$ was taken to be 15 km. For simplicity, it was assumed that

$G_{wave}[u,z]$ is determined only by the profile of $u$ at the equator and that it has a specified latitudinal structure $exp(-\frac{1}{2}(\varphi/10°)^2)$ to give equatorial confinement. In this type of simple wave forcing parametrization, inclusion of vertical diffusion of momentum is essential to allow an oscillation (see, e.g., Plumb 1977) and plays an important role close to the level where the wave forcing is imposed ($z_{launch}$ in the model presented here). The diffusion term has therefore been included in $G[u,z]$ with the diffusivity $\nu$ taken to be 2 m$^2$ s$^{-1}$. The diffusion term is not intended to represent any specific process, but

simply as an accepted modelling device required for this type of simple formulation of wave forces. $G_{friction}$ is imposed as a linear frictional term with timescale 1 day applied below $z = 3$ km and above $z = 60$ km, added to a linear friction with timescale 2000 days applied at all levels.

The equations above may be reduced to elliptic equations for $\frac{\partial u}{\partial t}$ and $\frac{\partial \langle T \rangle}{\partial t}$, where $\langle \cdot \rangle$ indicates the area-weighted latitudinal integral, in terms of $u$, $v$, $w$, $T$, $Q_{volc}, Q_{rad}$ , $G_{wave}$ and $G_{friction}$. Straightforward finite difference approximations to these

equations may then be solved using sparse matrix routines allowing the fields $u$ and $\langle T \rangle$ to be advanced in time. These fields are together sufficient to determine the entire $T$ field through (3). The results reported below were obtained with 120 levels in $z$ and 64 latitude points and with a time step of 0.1 day. It was verified that the results were robust to changes in these choices. If the aerosol heating $Q_{volc}$ and the wave force $G_{wave}$ are zero then the system simply relaxes towards the resting basic state.

Results from this model, discussed in Sect. 3.2, are used to support our arguments about the role of vertical velocity

perturbations driven by volcanic aerosol heating in the disruption of the QBO.

## 3. Results

### 3.1 The response of the QBO to a tropical volcanic eruption in the UM-UKCA model

### 3.1.1 Internal variability of the QBO in UM-UKCA

First, we evaluate the degree of natural variability in the simulated QBO in UM-UKCA for each control ensemble member

(Fig. 2). Below 10 hPa, the standard deviation between control ensemble members is rarely greater than 4 m s$^{-1}$ (Fig. 2, stippling), indicating that the strength of each phase does not vary substantially among ensemble members. For control





simulations initialised during a w-QBO state, the rate of descent of the westerly phase is similar for each ensemble member, as shown by the overlapping zero-wind lines for the first 12 months as the westerly phase descends (Fig. 2a). The zero-wind lines of the ensemble members begin to deviate from each other after ~12 months as the phase changes from westerly to easterly at different times for each ensemble member. For control simulations initiated during an e-QBO (Fig. 2b), the difference in timing of the easterly-to-westerly phase transition causes the zero-wind lines for each ensemble member to diverge by up to 4 months. To assess the effects of volcanic eruptions in Sect. 3.1.2, we consider phases that ascend rather than descend or changes to the w-QBO zonal winds that occur within the first 12 months to be disrupted.

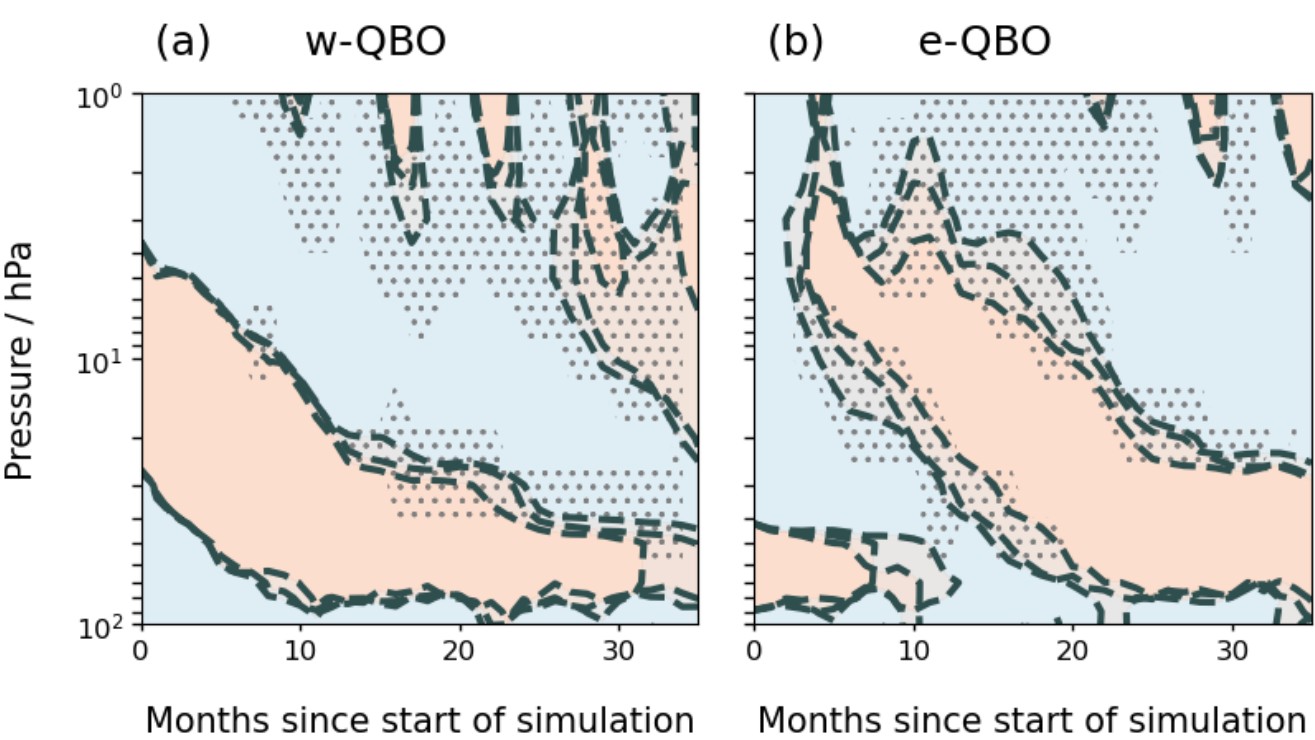

**Figure 2: Mean zonal wind direction averaged 2° N−2° S for all control ensemble member simulations initiated during (a) a w-QBO and (b) an e-QBO. Dashed lines indicate the zero-wind line for each ensemble member, orange shading indicates positive zonal wind, blue indicates negative zonal winds. Stippling occurs where the standard deviation of the zonal wind across control ensemble members is greater than 4 m s⁻¹.**

### 3.1.2 The response of the QBO progression to a tropical volcanic eruption in UM-UKCA

To evaluate any changes to the QBO after tropical volcanic eruptions, we present the zonal winds of the control ensemble mean and the zonal winds after each volcanic eruption in Fig. 3, and the zonal wind anomalies calculated compared to the



control ensemble mean in Fig. 4. We find that the progression of the e-QBO state is more strongly affected by a tropical volcanic eruption than the w-QBO. For a 60 Tg eruption initiated during a w-QBO state, the descent of the westerly phase is delayed by about 10 months after the eruption (compare Fig. 3a to Fig 3e) but for the same eruption initiated during an e-QBO,

the overlying easterly zonal winds actually ascend initially and remain above 30 hPa for the duration of the simulation (cf. Figs. 3b and 3f). The evolution of the e-QBO case no longer resembles the control ensemble simulations; the westerly phase originally at 65 hPa also ascends to 25 hPa for the 60 Tg eruptions and to 40 hPa for the 15 Tg eruptions (cf. Fig. 3b to Figs 3d and 3f). The larger disruption to the e-QBO relative to the w-QBO is also found in simulations started in January (see Supplementary Text S2).

For 15 Tg eruptions initiated during a w-QBO state (Fig. 3c), the disruption is most clearly seen in the zonal wind anomalies in Fig. 4a. The pattern of the zonal wind anomalies with respect to the control, is similar to that created by the 60 Tg eruption (cf. Figs 4a and 4c), which suggests there may be a small effect from the eruption. Within the first 10 months after the eruption, the 60 Tg eruption forces a weak easterly zonal wind anomaly of 15 m s$^{-1}$ at the easterly-to-westerly zero-wind line and a westerly wind anomaly at the westerly-to-easterly zero-wind line compared to the control ensemble mean (Fig. 4c). This is

consistent with a delay in the descent of the westerly phase. The easterly wind anomaly covers most of the range of altitude where westerlies are present in the control case during the first 10 months after the eruption, meaning the westerly phase at these altitudes has weakened during this period. A weaker westerly phase is also evident on inspection of Fig. 3c and 3e.

For the eruptions initiated during an e-QBO, the delayed westerly phase leads to a significant positive zonal wind anomaly with respect to the control case of up to 40 m s$^{-1}$ between 40 hPa and 5 hPa for 60 Tg eruptions (Fig. 4d) and up to 20 m s$^{-1}$

between 20 hPa and 40 hPa for 15 Tg eruptions (Fig. 4b). Compared to the zonal wind anomalies produced by eruptions initiated during w-QBO (Fig. 4a, 4c), the anomalies are much larger in magnitude.





**Zonal wind (2°N−2°S): w-QBO**     **Zonal wind (2°N−2°S): e-QBO**

(a) Control ensemble mean     (b) Control ensemble mean

(c) Eruption releasing 15 Tg of SO₂     (d) Eruption releasing 15 Tg of SO₂

(e) Eruption releasing 60 Tg of SO₂     (f) Eruption releasing 60 Tg of SO₂

Months since start of simulation     Months since start of simulation



**Figure 3: Filled contours: Mean zonal wind averaged 2° N−2° S. Contour lines: Positive (solid) and negative (dashed) mean zonal wind shear (the change in wind speed with height) at intervals of 0.0025 s⁻¹ averaged 2° N−2° S. A black solid contour indicates a wind shear of zero. The left column shows simulations initiated during westerly QBO shear at 30 hPa, and the right column shows simulations initiated during easterly shear at 30 hPa for (a), (b) the control ensemble mean (3 members), (c), (d) 15 Tg eruptions, (d), (e) 60 Tg eruptions. A white solid contour indicates the zero-wind line for comparison to Fig. 4. Red triangles indicate the approximate altitude of SO₂ injection.**








**Figure 4: Zonal mean zonal wind (2° N–2° S) anomaly compared to the control ensemble for (a), (c) 15 Tg eruptions and (b), (d) 60 Tg eruptions. The left column shows an eruption initiated during westerly QBO shear at 30 hPa and the right column shows eruptions initiated during easterly shear. Black solid lines outline the zero-wind line after each eruption. Red triangles indicate the eruption height. Stippling indicates where the zonal wind anomaly is not significantly different to variation among control ensemble members at the 5 % level using a Student's t-test.**

### 3.1.3 The latitudinal structure of the QBO in response to a volcanic eruption

The UM-UKCA simulations also suggest changes in the latitude-height structure of the QBO after an eruption. To compare to the control structure and to observations, Fig. 5 shows the latitudinal position of the maximum westerly zonal mean wind in the region 10 hPa to 70 hPa between latitudes 15° N and 15° S for each simulation. In this region, the westerly wind is dominated by the QBO. The westerly maximum therefore represents the centre of the QBO westerlies and gives an indication of the latitudinal structure of the QBO. The same analysis cannot be repeated for an easterly wind because QBO easterlies cannot easily be separated from the subtropical easterlies.

The control ensemble members follow similar paths and differ in latitude by 3° at most in our model simulations. They are consistently found within the 95th percentile of the latitudinal range from observations (Fig. 5, grey shading), and within 4° of the equator. Without perturbation from a large-magnitude tropical eruption, the position of the westerly maximum oscillates from approximately 4° N to 4° S about the equator in a 12 month cycle. The westerly maximum of the QBO is located in the hemisphere with the strongest extratropical westerlies, the winter hemisphere, creating a seasonal movement of the QBO westerlies driven by changes in zonal winds at higher latitudes.

After the eruptions initiated during a w-QBO state in July (Fig. 5a), the QBO westerly maximum deviates from the latitude of the control ensemble members and moves beyond the 95th percentile of the observed latitude range in the Northern Hemisphere. For the 60 Tg eruption, the QBO westerly maximum moves to 14° N and is still displaced relative to the controls 10 months after the eruption whereas for the 15 Tg eruption the excursion is smaller and shorter in duration. Sensitivity simulations performed in January show the westerly maximum moves in the opposite direction (reaching 12° S for a 60 Tg eruption) following an eruption initiated during a w-QBO state (Fig. S2a).

The latitudinal movement is smaller for eruptions initiated during an e-QBO compared to during a w-QBO. For the 60 Tg eruption, the westerly maximum moves further into the Southern Hemisphere compared to the control, but is only located beyond the 95th percentile of the range from observations for a single month (Fig. 5b). The 15 Tg eruption initiated during an e-QBO does not affect the latitudinal position of the westerly maximum. The movement into the Southern Hemisphere is in the opposite direction to the expected seasonal cycle and also opposite to the movement in response to an eruption initiated during a w-QBO. Comparing different seasons, the response of the e-QBO to an eruption in July is smaller than the response in January; both the 60 Tg and 15 Tg eruptions are displaced into the Southern Hemisphere after an eruption in January and the 60 Tg eruption causes the westerly phase to reach 8° S (Fig. S2b)





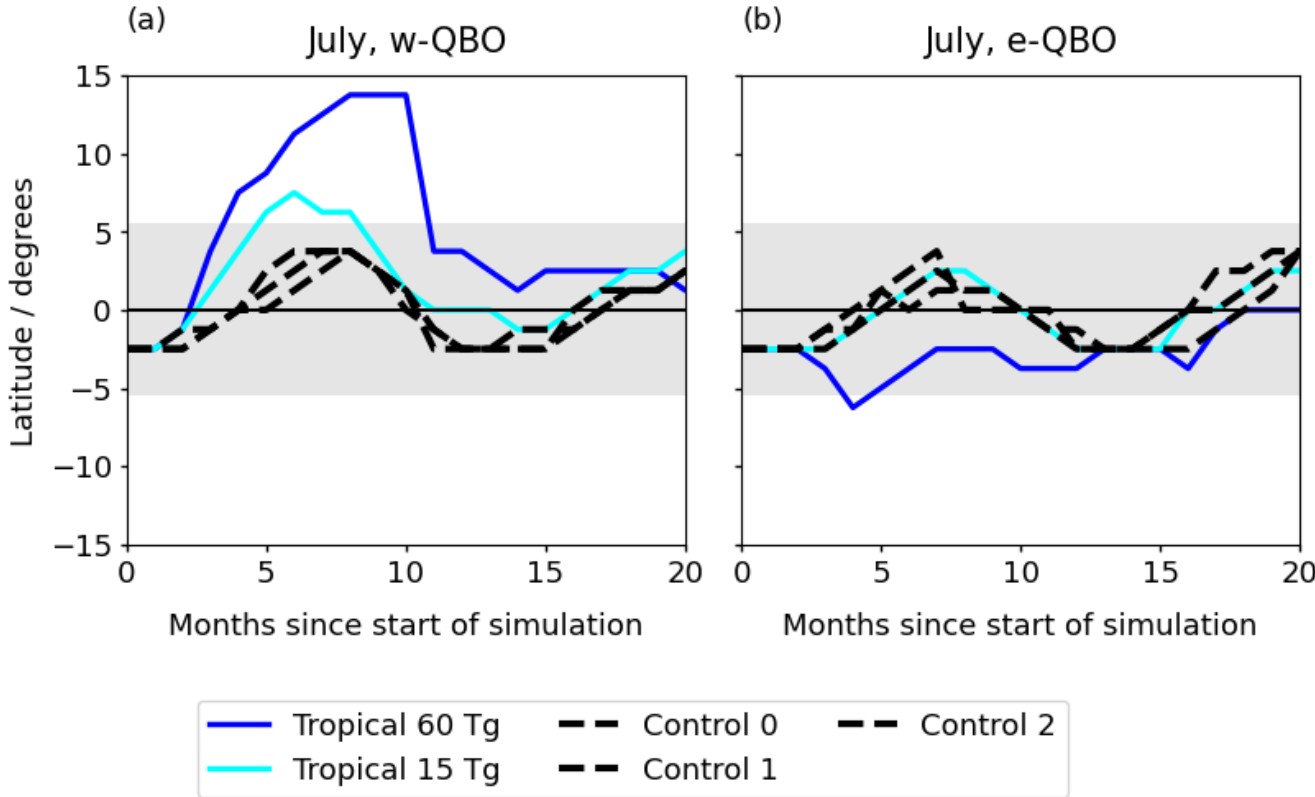

**Figure 5: Latitudinal structure of the QBO for (a) eruptions initiated during a w-QBO and (b) eruptions initiated during an e-QBO. Solid lines indicate latitudinal position of the QBO westerly maximum in the region 10 hPa–70 hPa, 15° N–15° S following 60 Tg and 15 Tg eruptions and black dashed lines indicate the latitudinal position of the QBO westerly maximum for each of the three control ensemble members. Grey shading covers the latitudes that contain 95% of the variation in the position of the observed QBO westerly maximum.**


The three observed large-magnitude eruptions (1963 Mt. Agung, 1982 El Chichón, 1991 Mt. Pinatubo) each took place during an e-QBO state and are closest in magnitude to the 15 Tg eruption, which shows no change from the control (Fig 5b, light blue line). Therefore, analysis of observations are currently not useful for verification of model results.

**3.2 Identifying the primary mechanism causing the QBO disruptions**

**3.2.1 Using the 2-D dynamical model**


The results of UM-UKCA suggest changes to the progression and latitudinal structure of the QBO that vary depending on the QBO state at the time of eruption. The model presented in Sect. 2.4 is now used to illustrate the effect of a simple representation



of aerosol heating on the evolution of the QBO. This will examine the change in the QBO progression following w-QBO and e-QBO states when both configurations are forced by an identical heating and allows assessment of the relative roles of the thermal wind balance and upwelling mechanisms.


Before discussing the impact of the heating perturbation on the QBO, it is useful to note the response to a heating perturbation in a resting atmosphere, without any wave forcing, i.e., setting $G_{wave} = 0$ in Eq. (2). The heating is confined in the latitude range [−10°, 10°] and in the height range [17.5 km, 27.5 km] and within this region takes a $\cos^2$ form in both latitude and height. The maximum value of the heating is 0.5 K day$^{-1}$. Figure 6a shows the height-latitude distribution of $Q_{volc}$. Also shown,

as dashed contours with the same contour interval, is the modification of $Q_{volc}$ by the term $-w\kappa T_b/H$ in Eq. (1), i.e., the adiabatic cooling/warming associated with ascent/descent. The magnitude of this term is substantially smaller and spread more broadly in latitude than that of $Q_{volc}$. Comparison with the temperature response (Figure 6b) confirms that the response to $Q_{volc}$ is strongly influenced by the the vertical velocity anomaly. Figure 6c shows the vertical velocity response, $w$, averaged over the 1 year for which the heating is applied. As expected from dynamical theory, the temperature response is latitudinally

broad compared to the heating because upwelling in the centre of the heating region leads to adiabatic cooling there, and downwelling at the edges of the heating region, and outside it, leads to adiabatic warming. This combination of adiabatic cooling and warming spreads out and reduces the temperature response.





(a)

(b)  (c)


**Figure 6: (a) Applied heating field $Q_{volc}$ (solid contours) and $Q_{volc} - w\,\kappa\,T_b/H$ (dashed contours). Contour interval is $10^{-6}$ K s$^{-1}$. (b) Temperature response to applied heating, averaged over the 1 year period for which the heating is applied. Contour interval is 0.5 K. (c) Corresponding response in vertical velocity. Contour interval is $10^{-4}$ m s$^{-1}$, positive contours solid, negative contours dashed, zero contour not shown.**

Turning now to the case where $G_{wave}$ is present, with the parameters specified in Sect. 2.4, the model generates a QBO with period of about 24 months. Figure 7a shows the time-height variation of $u$ at the equator. Note that the wave forcing operates only above 15 km; therefore, the variation of $u$ below 15 km is not very relevant to the QBO (the magnitude of the variation could be reduced by changing parameters in the model 'troposphere', but there are advantages to keeping the model formulation as simple as possible). Focusing on behaviour above 15 km it may be seen that the modelled QBO exhibits the





characteristic strong asymmetry in descent rates in the lower stratosphere between easterly shear zones and westerly shear zones, allowed by the inclusion of the vertical advection term in (2) and the thermal damping term in (1) (the latter statement has been verified by omitting the corresponding terms from the model equations, in which case the asymmetry between the easterly and westerly shear zones disappears).

The QBO of Fig. 7a is now perturbed by applying the aerosol heating, $Q_{volc}$, for a period of 1 year. Figures 7b and 7c show the time-height variation of $u$ when the aerosol heating $Q_{volc}$ is applied. The shading denotes the 1-year period when the heating is active in each case. In the first case (Fig. 7b), the beginning of the period of heating (24 months into the simulation) coincides with an e-QBO configuration. The effect can be measured by the delay to the phase transition of the QBO at the end of the heating period, which is about 14 months. In the second case (Fig. 6c), the beginning of the period of heating (38 months

into the simulation) coincides a w-QBO configuration. In this case the resulting delay to the phase transition of the QBO at the end of the heating period is about 6 months.



(a)

(b)                                            (c)

Figure 7: (a) Evolution of $u$ at equator in control experiment. Contour interval is 2.5 m s$^{-1}$, positive contours solid, negative contours dashed, zero contour not shown. (b) Evolution of $u$ when heating is applied for months 25–36 of the simulation, (i.e., an e-QBO configuration). These months are indicated by shading. (c) Evolution of $u$ when heating is applied for months 41–52 of the simulation (a w-QBO configuration).

The importance of the vertical advection of momentum by the vertical velocity response to $Q_{volc}$ in determining this delay can be illustrated by considering a modified case in which the vertical transport of momentum by the vertical velocity anomaly forced by the volcanic heating is removed, but the vertical velocity associated with the secondary circulation of the QBO is retained. (If the latter is also removed then the asymmetry in descent rates between easterly and westerly shear zones





disappears, confirming the conventional understanding of this asymmetry.) The details of this modified dynamics are set out in the Supplementary Text S1.


Figure 8 shows the time height variation of $u$ at the equator according to the modified dynamics, for heating $Q_{volc}$ applied during an easterly shear and a during westerly shear, corresponding respectively to the cases shown in Figures 7b and 7c. Comparing against Fig. 7a it may be seen that under the modified dynamics the effect of the heating (which operates only in the thermodynamic equation) on the QBO evolution is very small compared to that under the full dynamics; the shift in phase

of the QBO following the heating period in both cases is about 1 month, compared to 14 months or 6 months under standard dynamics. This provides clear evidence for the key role of the vertical advection of momentum due to the vertical velocity response to the heating in perturbing the evolution of the QBO. The direct modification of temperature by the applied heating, as implied to be crucial in the 'thermal wind mechanism', appears to be of secondary importance.

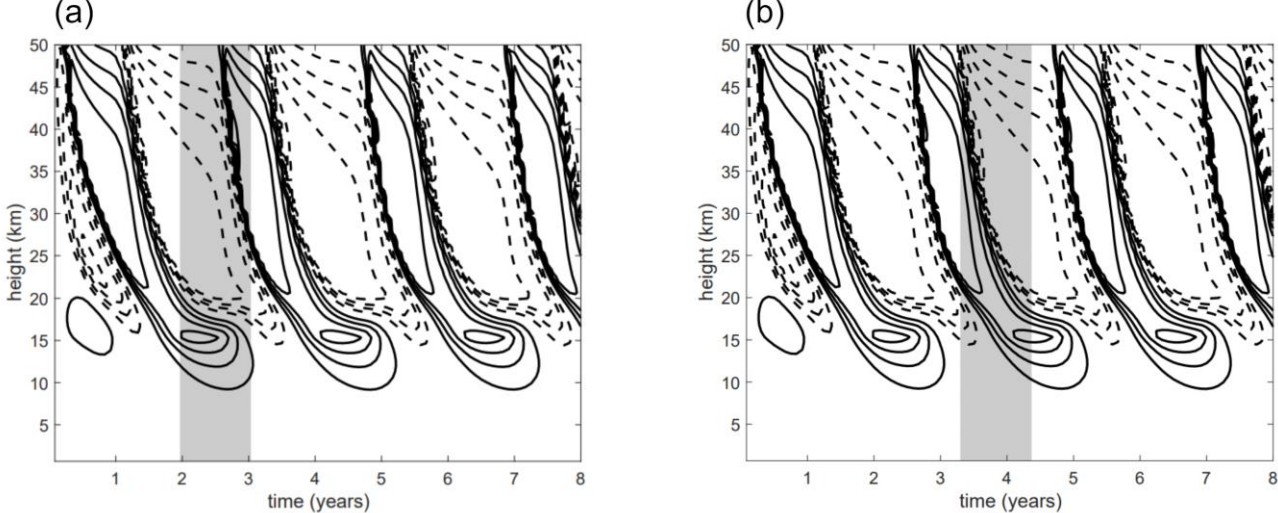


**Figure 8: Evolution of $u$ at equator in experiments with QBO perturbed by heating under modified dynamics. Contour interval is 2.5 m s⁻¹, positive contours solid, negative contours dashed, zero contour not shown. (a) Evolution of $u$ when heating is applied for months 25−36 of the simulation. These months are indicated by shading. Corresponding evolution of $u$ under standard dynamics is shown in Figure 7b. (b) Evolution of $u$ when heating is applied for months 41−52 of the simulation. Corresponding evolution of $u$**
**under standard dynamics is shown in Figure 7c.**

### 3.2.2 Using UM-UKCA

In our UM-UKCA simulations, $w$ increases predominantly close to the equator in response to a volcanic eruption. The temperature anomaly due to the eruption is broad, in contrast to $w$ and the aerosol anomaly, for which the largest values are latitudinally confined (Fig. 9a, 9b for 60 Tg eruptions, Fig. S7, S6 for 15 Tg eruptions). As seen schematically in Fig. 1c and

using the 2-D model in Fig. 6, this indicates that most of the aerosol heating drives anomalies in $w$ rather than increasing the





temperature. As in Sect. 3.2.1, we attribute the delay in the descent of the QBO phases in UM-UKCA to the vertical advection of momentum from the positive $w$ driven by the aerosol heating.

In UM-UKCA, the $w$ anomaly forced by the aerosol heating from both 60 Tg and 15 Tg eruptions results in a complete halt

of the descent of the QBO when the model is initialized in the e-QBO state but not when it is initialized in the w-QBO state (since the the QBO secondary circulation for the easterly shear zone augments the aerosol effect on the upwelling, but for the westerly shear zone diminishes it). In fact, for an eruption initiated during an e-QBO state, we find that the QBO phases move up in altitude.






**Figure 9: Two months after 60 Tg eruptions in July: (a), (b) shows the _w_ anomalies (filled contours), positive temperature anomalies (black solid contours) and sulfate mass mixing ratio anomalies (coloured line contours) with respect to the control ensemble mean. The sulfate mass mixing ratio contours are at interval of 5 x10$^{-8}$, 15 x10$^{-8}$, 50 x10$^{-8}$ kg kg$^{-1}$ and the temperature contours are in intervals of 1.5 K. (c), (d) The monthly mean zonal wind after the eruption (filled contours) compared to the control (black line contours). The filled contour intervals are the same as the line contours, with dashed lines representing negative zonal winds in the line contours. Panels (a) and (c) show results for the w-QBO, and panels (b) and (d), show results for the e-QBO.**

### 3.3 The aerosol distribution and possible effects on the QBO response

Above, we show that the QBO progression close to the equator is perturbed in similar ways in the 2-D and 3-D models, and confirm that the disruption to the QBO can be explained primarily by the increase in upwelling caused by the eruption on the QBO momentum balance. Importantly, the difference in response between the w-QBO and e-QBO states can be explained by considering the effect of the upwelling induced by aerosol heating on the existing QBO dynamics, without the need to consider differences in the forcing between QBO states. However, we also identify additional changes to the QBO latitudinal structure in UM-UKCA (Sect. 3.1.3), which were dependent on QBO state and season. We now explore whether the latitudinal structure of the QBO could be influenced by the distribution of aerosol and subsequent forcing. Here we analyse the circulation changes to the stratosphere in the latitude-height plane following each QBO state using the 60 Tg July eruption scenarios, although the same pattern of results can be identified for the 15 Tg eruptions, which are shown in the Supplementary Text S3.

To establish the relationship between the volcanic sulfate aerosol distribution and the temperature anomalies, Figs. 9 and 10 compare the latitudinal structure of the zonal wind, temperature anomaly and sulfate aerosol anomaly for months two and five after an eruption. Two months after the eruption is the first month with a volcanic sulfate aerosol burden in the contour range chosen (5 x10$^{-8}$ kg kg$^{-1}$) and five months after eruption the volcanic sulfate aerosol burden is at its highest.

Two months after the July eruption, in September, transport of aerosol into the winter hemisphere (Southern Hemisphere) is enhanced in the w-QBO case presumably due to Rossby wave breaking and the ability of Rossby waves to propagate between the subtropical westerlies and the westerly phase of the QBO (Fig. 9a). On the other hand, wave propagation and breaking is prohibited in the e-QBO case due to blocking by easterly winds (Fig. 9b). This results in more sulfate aerosol entering the Southern Hemisphere after an eruption initiated during a w-QBO compared to a symmetrical distribution after an eruption initiated during an e-QBO. A similar pattern is found for 15 Tg eruptions (Fig. S7), although the magnitude of the temperature and sulfate anomalies are smaller. For eruptions in January, sulfate aerosol is transported into the Northern Hemisphere after an eruption initiated during a w-QBO (Fig. S3).

Five months after the eruption, in December, the aerosol burden is still highest in the Southern Hemisphere after an eruption initiated during a w-QBO, and there is clearly limited transport into the Northern Hemisphere (Fig. 10a) despite the seasonal change to westerlies supporting Rossby wave propagation there. The QBO westerly phase has become broad and developed a hemispheric asymmetery with its peak winds moved off the equator into the Northern Hemisphere (centered near 10° N)





whereas the control westerly peaks much closer to the equator (Fig. 10c). For the 15 Tg eruption, the westerly phase is centered near 7° N (Fig. S8c).

Whilst the direct modification of the temperature structure by the aerosol heating (the 'thermal wind balance mechanism') is likely to be very ineffective in initially perturbing the progression of the QBO in the time-height plane, it may be more important in affecting secondary characteristics at longer time scales such as the latitudinal structure of the zonal wind away from the equator. For an eruption initiated during a w-QBO, the reduced transport of volcanic sulfate aerosol into the Northern Hemisphere during July−September creates a steep aerosol gradient. This also allows a steep heating and temperature gradient to form over time, consistent with an enhanced westerly shear zone in thermal wind balance in the Northern Hemisphere. Upon the arrival of Northern Hemisphere winter (December), the westerly phase shifts into the Northern Hemisphere and becomes stronger than the unperturbed cases at these latitudes (Fig 10c, Fig. S8c). The enhanced meridional potential vorticity gradient associated with this off-equatorial westerly phase may effectively act as a transport barrier to the sulfate thereby reinforcing the steep meridional aerosol gradient and confining it more to the easterly phase (see Fig. 10a and 10c).

For an eruption initiated during an e-QBO state, the volcanic sulfate aerosol burden is more evenly distributed in each hemisphere compared to the w-QBO case for 60 Tg and 15 Tg eruptions (Fig. 10b and Fig. S8b, respectively) although for the 60 Tg eruption the burden is slightly higher in the Northern Hemisphere.




**w-QBO**   **e-QBO**

**Figure 10: Five months after 60 Tg eruptions in July: (a), (b) shows the *w* anomalies (filled contours), positive temperature anomalies (black solid contours) and sulfate mass mixing ratio anomalies (coloured contours) with respect to the control ensemble mean. The sulfate mass mixing ratio contours are at interval of 5 x10⁻⁸, 15 x10⁻⁸, 50 x10⁻⁸ kg kg⁻¹ and the temperature contours are in intervals of 1.5 K. (c), (d) The monthly mean zonal wind after the eruption (filled contours) compared to the control (black contours). The filled contour intervals are the same as the line contours, with dashed lines representing negative zonal winds in the line contours. Panels (a) and (c) show results for the w-QBO, and panels (b) and (d), for the e-QBO.**

## 4. Discussion

Using the interactive aerosol model UM-UKCA, we find that, in response to tropical volcanic eruptions releasing either 15 Tg or 60 Tg of $SO_2$, downward propagation of the QBO phases during e-QBO initial conditions is delayed more than during w-QBO conditions (Fig. 3), and that the response is consistent regardless of season of eruption (Fig. S1). In fact, the westerlies




in the lower stratosphere temporarily ascend after an eruption during an e-QBO. Using a simple 2-D model of the QBO we show that these changes can be explained by considering the change in vertical velocity triggered by the eruption; a heating applied at the equator causes a vertical velocity which slows the descent of the QBO phases by its effect on vertical momentum advection ($w\frac{\partial u}{\partial z}$, Equation 2). Recent papers have suggested that two distinct mechanisms, the 'thermal wind balance mechanism' and the 'upwelling mechanism' play a role. Our conclusion is that the the latter, the 'upwelling mechanism' is

much more important and provides a clear explanation for the dependence of the response on QBO phase and on eruption magnitude. In particular, variation in response between each QBO state is due to the fact that, in the unperturbed QBO, the easterly shear zone propagates downward more slowly than the westerly shear zone, because of the QBO secondary circulation. Therefore, the addition of an upwelling of a given magnitude, determined by the amount and location of the volcanic aerosol, has a stronger effect on the propagation of the easterly shear zone than on that of the westerly shear zone (as summarised

schematically in Fig. 11a). The possibility of an upwelling mechanism for the disruption of the QBO by volcanic eruptions and the larger potential effect on easterly shear zones was previously suggested by Dunkerton (1983) and illustrated in a 1-D model, which  seems to have been overlooked in more recent work on effects of volcanic aerosol and of sulfate geoengineering on the QBO. The 1-D model is restrictive since it allows no alternative to the upwelling mechanism for QBO disruption, but we have shown here that the upwelling mechanism remains the dominant effect in 2-D and 3-D simulations.

In observations, the QBO westerlies after the 1991 Mt. Pinatubo eruption were unusually extended in the lower stratosphere, which is noted by Labitzke (1994) although not specifically attributed to the eruption, as were the QBO westerlies after the earlier 1963 Mt. Agung eruption, which Dunkerton (1983) suggests was due to the eruption. In our study, for the 15 Tg eruption initiated during an e-QBO (the simulation that resembles the 1991 Mt. Pinatubo eruption most closely), an extension of the westerly phase at 50 hPa occurs, but we also find that the eruption forces the westerly phase to move upwards in altitude rather

than just slowing the descent. This may be due to differences between our simulation and the specific conditions of the observed eruptions or, perhaps more likely as we suggest below, that the QBO in UM-UKCA is more sensitive to an eruption than the observed QBO due its longer period.

    Our results agree with DallaSanta et al. (2021) in that the initial QBO state strongly affects the QBO response to forcing from

a Pinatubo-magnitude or Tambora-magnitude volcanic eruption, and that the e-QBO is most strongly affected. DallaSanta et al. (2021) characterise this by an increase in the period relative to climatology following an eruption during an e-QBO state. They also find that an eruption at the onset of formation of upper stratospheric westerlies causes the period of the QBO to decrease, and go on to conclude that the period of the QBO will decrease for an eruption initiated during a w-QBO state, i.e., 'the QBO will hasten towards a state with lower stratospheric westerlies'. However, we find a delay in the descent of the

westerly phase for eruptions initiated during a w-QBO in our simulations, which does not fit the concluding statement by DallaSanta et al. (2021). On closer inspection of the findings by DallaSanta et al. (2021), they only record a decrease in the period for QBO states that contain both upper and lower stratospheric westerlies (and an easterly phase in between). We did





not test this QBO state. Given that there is a large amount of natural variation in the QBO period and that it may be affected by atmospheric changes occurring over several months, it is unclear how the decrease in period identified by DallaSanta et al.

(2021) manifests itself and what the driving mechanism is.

The model-simulated QBO disruptions caused by Pinatubo- and Tambora-magnitude volcanic eruptions that we have discussed here occur on time scales shorter than or comparable to the QBO period and we expect them to be temporary in nature (though the short duration of the simulations has not allowed that to be confirmed explicitly). In this sense, the behaviour of the QBO

after a volcanic eruption is analogous to the recent observed dynamical disruptions caused by temporarily enhanced extratropical wave activity (Anstey et al. 2021) after which the QBO resumed normal behaviour, and the disruption may be more clearly characterised by considering the phase evolution rather than yearly means or changes to period length. The formation of dominant easterlies predicted by Brenna et al. (2021) were not seen in our study, due to differences in the magnitude of the simulated eruption. From our results, which show the QBO phases ascending after an eruption during an e-

QBO, it is conceivable that the increase in $w$ after a 1000 Tg eruption is large enough that the QBO is completely removed, leaving easterlies.

Most studies focussing on sulfate aerosol geoengineering have not separated the effects of vertical advection of momentum, $w \frac{\partial u}{\partial z}$, from the direct modification of the temperature structure by aerosol heating coupled to thermal wind balance. Earlier

studies (e.g., Aquila et al.; 2014) attribute the slower descent of the QBO phases to an increase in $w$ from the sulfate heating, but the greater effect on the e-QBO is attributed to the thermal wind relation not the dynamical properties of the easterly shear zone itself. Over the longer timescales studied in geoengineering simulations, the heating profile is likely to broaden as the aerosol is dispersed, as found by Franke et al. (2020), and a broader heating anomaly leads to a smaller increase in tropical $w$. These factors, coupled with feedbacks from changes in the extra-tropics makes attribution of drivers over the long timescales

of geoengineering more challenging. However, our study suggests that the preference for sulfate geoengineering to halt the QBO following e-QBO conditions, that is, in a state where the QBO phase is easterly in the mid-stratosphere and westerly in the lower stratosphere, is due to an increase in $w$, combined with the QBO secondary circulation which results in a slower descent of the easterly shear zone compared to the westerly shear zone.

We propose that the primary effect of a stratospheric heating on the QBO can be assessed by considering: (A) the change in tropical upwelling close to the equator and (B) the implication of the upwelling for the QBO momentum balance. Considering (A), the magnitude of the change in upwelling will vary with injection protocol (e.g., Haywood et al., 2022) and model differences (Niemeier et al., 2020). Broader heating anomalies and heating anomalies away from the equator will decrease the magnitude of the upwelling close to the equator and may even lead to a downwelling. Therefore, regional injection strategies

and higher latitude injections (e.g., Richter et al., 2017; Franke et al., 2021) show weaker QBO responses. A downwelling





anomaly at the equator could lead to a more rapid downward propagation of the QBO and a decrease in the QBO period. Indeed, there is a hint in Fig. 12 of Franke et al. (2021) (compare a and b with c and d) that the 2-point injection of sulfate has caused a slight reduction in QBO period.

Considering (B), the ability of the change in upwelling to disrupt the QBO would depend on the QBO momentum balance in each model. In our study, the momentum balance associated with each QBO state caused the e-QBO initial condition to be more affected than the w-QBO. However, the QBO momentum balance also varies among models; the unperturbed QBO simulated in UM-UKCA descends more slowly than the observed QBO, so it is likely to be more sensitive to increased upwelling due to aerosol heating. Sulfate geoengineering studies have previously found that model variables including BDC

strength and gravity wave parameterisations, which affect the evolution of the QBO, lead to differences among model responses. In particular, Franke et al. (2021) find that a QBO with longer lower stratospheric easterly phases was locked into a permanent easterly phase, whereas a QBO with longer lower stratospheric westerlies (as in observations) was locked into a permanent westerly phase in response to continuous sulfate injection.

Given the highly variable conditions of volcanic eruptions including variations in injection height (Aquila et al. 2014; Tilmes et al. 2018), injection latitude, sulfate mass and eruption duration, it is likely that an even greater variety of changes to the QBO are possible than are explored in our study. The predictability of these responses based on the change in vertical velocity at the equator ought to be investigated further.





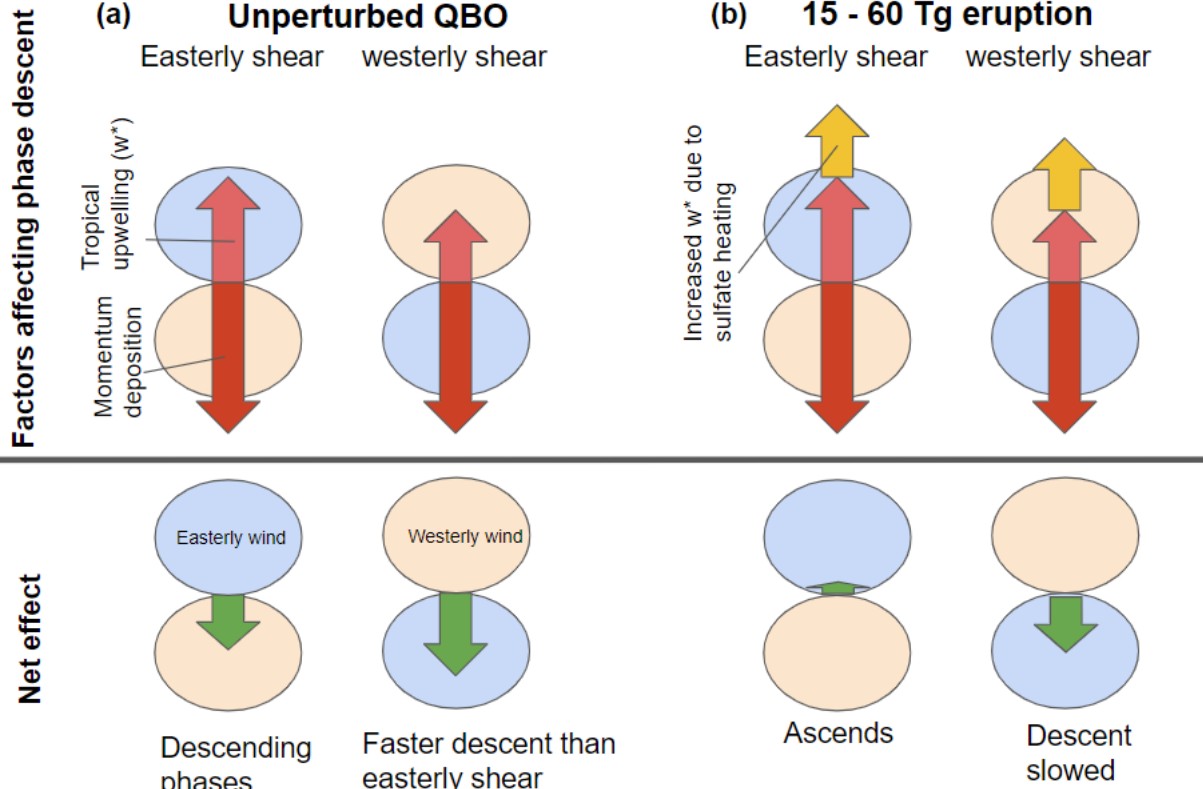

**Figure 11: Schematic representing (top row) the most important factors controlling the QBO momentum balance and (bottom row) the net effect of these factors on the easterly shear descent rate and the westerly shear descent rate during (a) an unperturbed QBO and (b) a Tambora-magnitude or Pinatubo-magnitude eruption simulated in the tropics. Arrows indicate the effect on the movement of the QBO phases due to mean vertical advection of zonal momentum, $w\frac{\partial u}{\partial z}$ (pink), momentum deposition (red), the change in $w\frac{\partial u}{\partial z}$ due to the eruption (yellow) and the net effect of these (green).**

The UM-UKCA simulations, which include feedbacks among sulfate aerosol distribution, atmospheric dynamics and heating also show changes in the latitudinal structure of the QBO. For eruptions initiated during a w-QBO state especially, there is an asymmetric distribution of volcanic sulfate aerosol between the hemispheres due to seasonal variation in BDC strength (Tilmes et al., 2017) and we find this has an impact on the latitudinal structure of the QBO. We find that the westerly phase of the QBO becomes broader and moves into the hemisphere with the lowest volcanic sulfate mass burden, the summer hemisphere (Fig. 5a). For 60 Tg eruptions simulated using UM-UKCA, the maximum QBO westerly is predicted to be located beyond ± 10°, with the direction dependent on the season of the eruption (Southern Hemisphere for a July eruption, Northern Hemisphere for a January eruption). This creates a temporary weakening of the QBO westerly phase at the equator as it moves off equator (shown by a negative zonal wind anomaly in Fig. 4a, 4c). The QBO westerly phase has never been observed to move to this latitude, and the consequences are unexplored. A possible indication of this phenomenon in a different climate model is identified by Richter et al. (2017). They show easterly and westerly anomalies in the average QBO in response to an





asymmetrical sulfate geoengineering experiment, with the westerly anomaly occurring in the opposite hemisphere to the injection, although they do not go on to show whether the anomaly was caused by movement of the QBO westerly phase.


Furthermore, we suggest that there is a feedback effect between the movement of the QBO westerly phase and the transport of aerosol that allows the westerly phase to move unusually far away from the equator for several months for eruptions initiated during a w-QBO state. For an eruption initiated in July, easterlies on the northern flank of the QBO during late summer to early autumn prevent wave-driven dispersion of volcanic aerosol into the Northern Hemisphere, which creates a strong

meridional gradient in aerosol and, hence, in heating. During a w-QBO state, this strong meridional heating gradient aligns with the westerly phase of the QBO and supports its location slightly to the north of the equator. As the season progresses into early Northern Hemisphere winter, the westerly phase, which is now off the equator, may then act as an effective transport barrier further preventing wave-driven mixing and sustaining the meridional aerosol and westerly phase off the equator. Indeed, for 60 Tg eruptions initiated during a w-QBO in July, aerosol is still mainly found in the Southern Hemisphere and the aerosol

circulation appears to be inhibited by the position of the westerly phase at 10° N (Fig. 10c). Since the seasonal circulation is opposite between January and July, eruptions initiated during a w-QBO state in January force the westerly phase to strengthen in the Southern Hemisphere (Fig. S4). The same changes are identified in the 15 Tg eruptions at a smaller magnitude (Fig. S8).

**6. Conclusions**

We have investigated the changes to the time-evolving progression and latitudinal structure of the QBO in response to a tropical volcanic injection of SO$_2$ into the stratosphere using a 2-D model and an interactive stratospheric aerosol model (UM-UKCA).

Our simulations show that after tropical volcanic eruptions injecting 60 Tg or 15 Tg of SO$_2$, the QBO phases can be delayed and even ascend in altitude. We find that the state of the QBO (defined here by the sign of the vertical shear at 30 hPa) is critical in determining the response of the QBO to a volcanic eruption, and that the e-QBO state is most affected. The disruption

to the QBO progression is the primary QBO response to the eruption, and the e-QBO shows a larger response than the w-QBO for eruptions initiated in both July and January. This is consistent with the hypothesis that an increased tropical upwelling (and hence vertical advection of zonal-mean momentum) is the most important contributor to the QBO response; the QBO secondary circulation causes easterly shear zones to descend faster than westerly shear zones, so that the additional upwelling driven by aerosol heating has a stronger effect on easterly shear zones. However, we acknowledge that the QBO produced by

UM-UKCA descends more slowly than the observed QBO and therefore may be more sensitive to disruption than a more realistic QBO. To increase confidence as to which magnitudes of eruption may disrupt the QBO, further simulations are needed using different climate models for a range of possible eruptions and different QBOs. Furthermore, the magnitude of the

disruption may be sensitive to the injection height of $SO_2$ relative to the phase of the QBO; Marshall et al. (2019) previously showed that injection height controls radiative forcing from tropical eruptions, and this should be an area of future work.

We also find that feedbacks due to transport of the aerosol cause the QBO to move further into the hemisphere with the lowest volcanic sulfate aerosol burden, and become broader. This effect was sensitive to the season and the state of the QBO since these factors influence the evolution of the sulfate aerosol, and the w-QBO showed the greatest latitudinal movement in our simulations. Seasonal differences in transport are especially pronounced during the w-QBO state so that movement occurs into the Northern Hemisphere after an eruption in July and into the Southern Hemisphere after and eruption in January. The

latitudinal movement of the QBO westerly phase could have implications for where the strongest pole to mid-latitude mixing occurs, affecting ozone concentrations (Diallo et al., 2018) and the distribution of the volcanic sulfate aerosol itself.

Lower stratospheric westerlies, such as those simulated after an eruption initiated during an e-QBO state, can strengthen the polar vortex and affect tropospheric weather (Kidston et al., 2015; Gray et al., 2018; Yamazaki et al., 2020). As the QBO increases long-term interannual predictive skill for Northern Hemisphere weather forecasting via teleconnections, disruptions

reduce the accuracy of predictions (Scaife et al., 2014; Anstey et al., 2021). Additionally, species such as ozone and stratospheric water vapour have different signals depending on the zonal wind shear and their distribution is important for climate change projections. The impacts from previous observed disruptions (Newman et al 2016; Osprey et al., 2016; Hitchcock et al., 2018; Anstey et al., 2021) are poorly quantified despite an increased probability of a QBO disruption in a warmer future climate so a greater understanding of these impacts is necessary to increase societal resilience to climate change

and volcanic eruptions.

**Code/data availability**

The data is available upon request to the corresponding author. Python 3.7 code is available on the GitHub repository of the corresponding author. Please get in touch for further information.

**Author contributions**

Conceptualization: FB, LM, AS

Supervision: LM, AS

Investigation & Methodology: LM (UM-UKCA simulations), PH (2-D model simulations)

Data Curation: LM

Anaysis and visualisation: FB

Writing: all authors



**Competing interests**

The authors declare there are no competing interests.

**Acknowledgements**

The authors would like to thank Luke Abraham for technical advice and useful discussions and Scott Osprey for useful discussions. Flossie Brown was funded by the NERC GW4+ DTP (award no. NE/S007504/1) and the Met Office on a CASE studentship. LM and AS acknowledge funding from NERC grant NE/S000887/1 (VOL-CLIM). This work used the ARCHER UK National Supercomputing Service (http://www.archer.ac.uk).

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
