# Peer review of "On the magnitude and sensitivity of the QBO response to a tropical volcanic eruption"

_EGUsphere, 2022_

## Author Comment (AC1)

**Reviewer response**

**We would like to thank the reviewers for their time, feedback and useful comments. The suggested modifications have certainly improved the manuscript.**

**In this document, the review comments are shown in blue. The author response is shown in *italics*. Edits to the manuscript are included in "quotation marks" with tracked changes.**

**REVIEW 1:**

L 68: "Sulfate geoengineering is, in effect, equivalent to a sustained volcanic eruption, although somewhat different to a short-lived, explosive eruption" A bit of a convoluted phrase. "somewhat different" – how?

*We have simplified this sentence to improve readability. In the following paragraph beginning on L87 we go into more details on the differences between sustained and short-lived eruptions.*

"There have been several recent studies investigating the response of the QBO to  solar radiation management (SRM) through artificial injection of SO₂ or sulfate into the stratosphere  , which can be considered as equivalent to a sustained volcanic eruption (although not necessarily equivalent to the short-lived, explosive eruptions in this study)."

20      L 108: Also see Pitari et al. (2016) which discusses this exact issue in the context of the four main volcanic eruptions.

*Thank you for this reference, we have included an additional sentence.*

"The time-evolving dispersion of the aerosol is likely to influence the forcing, and therefore the circulation patterns. For example, Pitari et al. (2016) find that a Pinatubo-magnitude eruption during e-QBO conditions delays meridional transport and increases e-folding times compared to an eruption during w-QBO conditions. This could lead to differences in the forcing and response between eruptions initiated during different QBO states and seasons"

L 130: "Inacted"?

30      *This typo has been corrected to 'Enacted'*

S2.2 Interested here as to why the authors don't reference or compare their results with those from Dhomse et al. (2020)? It seems like the model is the same, maybe a slightly different version, however the amount of SO2 is different, and Dhomse found a better agreement for lower injections of SO2. Their simulations are also AMIP-style. This needs to be discussed (and perhaps, if the models' versions are compatible, one could look at those already available simulations which are part of ISA-MIP and compare the QBO response…). In terms of the location of the injection (in altitude), the authors can also refer to the range explored in the ISA-MIP experiments (Quaglia et al.,

40

2022) – at least acknowledging that different altitudes of injection produce different clouds with, potentially, different QBO effects.

*Although comparison with other models and experimental designs would certainly be interesting, we feel it may complicate the messages already present in the paper. Since the eruption location, magnitude, height and model set up are different, a full comparison could become very complicated. As for a comparison with Dhomse et al. 2020, the models are different versions (UM vn8.4 vs. UM vn11.2) and the aerosol scheme is also slightly different (no meteoric smoke particles, and no dependency of the sulphuric acid condensation on the vapour pressure deficit, for example), aligning instead to that used in the UK Earth System Model (Mulcahy et al., 2020). Therefore, a straightforward comparison would be difficult with a major confounder being the slightly different eruption location (in terms of latitude and altitude) that would certainly affect the aerosol distribution. Our injections are representative of the eruptions of 1991 Mt. Pinatubo-magnitude and 1815 Mt. Tambora-magnitude but we do not seek to replicate them specifically.*

*Mulcahy, J. P., Johnson, C., Jones, C. G., Povey, A. C., Scott, C. E., Sellar, A., Turnock, S. T., Woodhouse, M. T., Abraham, N. L., Andrews, M. B., Bellouin, N., Browse, J., Carslaw, K. S., Dalvi, M., Folberth, G. A., Glover, M., Grosvenor, D. P., Hardacre, C., Hill, R., Johnson, B., Jones, A., Kipling, Z., Mann, G., Mollard, J., O'Connor, F. M., Palmiéri, J., Reddington, C., Rumbold, S. T., Richardson, M., Schutgens, N. A. J., Stier, P., Stringer, M., Tang, Y., Walton, J., Woodward, S., and Yool, A.: Description and evaluation of aerosol in UKESM1 and HadGEM3-GC3.1 CMIP6 historical simulations, Geosci. Model Dev., 13, 6383–6423, https://doi.org/10.5194/gmd-13-6383-2020, 2020.*

L. 433 Stop is inside the parenthesis but should be outside.

*This has been corrected.*

Figure 9 and 10 are really, really hard to interpret due to the different contours. I would suggest duplicating the figure for T and for SO4 mass rather than try to cram everything in one panel.

*Thank you for your feedback. The figures have been replaced in the main text and supplementary with 6 panels. References to panels in the text have been changed accordingly. Figure 9 is shown below with the updated figure caption.*

[Figure]

**Figure 9:** Two months after 60 Tg eruptions in July: (a), (b) shows the *w* anomalies (filled contours) and sulfate mass mixing ratio anomalies (coloured line contours) with respect to the control ensemble mean and (c), (d) shows the *w* anomalies (filled contours) and positive temperature anomalies (black solid contours).. The sulfate mass mixing ratio contours are at interval of 5 x10⁻⁸, 15 x10⁻⁸, 50 x10⁻⁸ kg kg⁻¹ and the temperature contours are in intervals of 1.5 K. Panels (e), (f) show the monthly mean zonal wind after the eruption (filled contours) compared to the control (black line contours). The filled contour intervals are the same as the line contours, with dashed lines representing negative zonal winds in the line contours. Panels (a) , (c) and (e) show results for the w-QBO, and panels (b) (d) and (f), show results for the e-QBO.

Overall, a panel showing AOD (both global over time, or 30N-30S, and latitudinal) could be very useful to understand better how the QBO is affecting the aerosols.

*We agree these are useful figures to include and the figures below have been added to the supplementary material. We reference these figures in the main text in the following places:*

L508: "Five months after the eruption, in December, the aerosol burden is still highest in the Southern Hemisphere after an eruption initiated during a w-QBO, and there is clearly limited transport into the Northern Hemisphere (Fig. 10a, Fig. S4b) despite the seasonal change to westerlies supporting Rossby wave propagation there. In fact, the sAOD remains higher in the Southern Hemisphere for the full simulation (c.f. Fig. S3b and S3c)."

L530: "For an eruption initiated during an e-QBO state, the volcanic sulfate aerosol burden is more evenly distributed in each hemisphere compared to the w-QBO case for 60 Tg and 15 Tg eruptions (Fig. 10b and Fig. S108b, respectively) although for the 60 Tg eruption the burden is slightly higher in the subtropical Northern Hemisphere. This can be confirmed in the sAOD at 5 months after the eruption (Fig. S4)."

*We also updated the supplementary text:*

"To further understand the changes in the QBO latitudinal structure, a comparison of latitudinal variation in sAOD 5 months after an eruption is shown in Fig. S4. We confirm that for an eruption during a w-QBO in January there is a steep decline in sAOD in the Southern Hemisphere, opposite to the sAOD pattern seen for an equivalent eruption in July. We also confirm that the sAOD for eruptions initiated during an e-QBO are more symmetrical than for the w-QBO. Finally, we show that for eruptions initiated in January, the e-QBO and w-QBO sAOD pattern are more similar than for the corresponding eruptions initiated in July. This may explain why the latitudinal movement predicted for eruptions in January are similar in magnitude and direction whereas for eruptions in July there is only a strong response seen during w-QBO initial conditions (Fig. 5). Further ensemble members with a variety of e-QBO states would be needed to confirm if the differences between the e-QBO latitudinal response in January and July is due to the season or natural variability among e-QBO states."

[Figure]

**Figure S3: SAOD at 550 nm over time for (a) the tropics (30° N – 30° S), (b) the Southern Hemisphere (30° S – 90° S) and (c) the Northern hemisphere (30° N – 90° N).**

[Figure]

**Figure S4: A latitudinal cross-section of sAOD at 550 nm 5 months after an eruption for each set of initial conditions for (a) 15 Tg eruptions and (b) 60 Tg eruptions.**

L 578 (paragraph) I agree with the authors' assessment that it is hard with a continuous perturbation to make similar attributions. However, I would stress here that "the preference for sulfate geoengineering to halt the QBO following e-QBO conditions" only refers to SG applied at the equator (see the cited Richter et al., 2017 and Kravitz et al., 2019), and not even in all climate models (see Niemier et al., 2020, but also Jones et al., 2022, Fig. 8).

*We have amended the paragraph to make it clear we refer to equatorial injection, since this is comparable with our own study. We have also clarified that the QBO does not shut down in all SG conditions or all models, but that in general, a tendency towards a QBO state with lower stratospheric westerlies is very common. Although there is considerable uncertainty about the forcing required to shut down the QBO, most models will shut down when the forcing is large enough. Jones et al. 2022 show complete shut down in 3/5 climate models and an increase in the QBO period, specifically lower stratospheric westerlies, in another climate model. Similarly, both the models compared by Niemeier et al. 2020 show a QBO shut down although one model does not shut down below 8 Tg year$^{-1}$ SO$_2$. Other injection strategies are discussed in the following paragraph.*

"Most studies focussing on sulfate aerosol  geoengineering have not separated the effects of vertical advection of momentum, $w\frac{\partial u}{\partial z}$, from the direct modification of the temperature structure by aerosol heating coupled to thermal wind balance. Earlier studies following an equatorial injection (e.g., Aquila et al., 2014) attribute the slower descent of the QBO phases to an increase in $w$ from the sulfate heating, but the greater effect on the e-QBO is attributed to the thermal wind relation not the dynamical properties of the easterly shear zone itself. Over the longer timescales studied in  geoengineering simulations, the heating profile is likely to broaden as the aerosol is dispersed, as found by Franke et al. (2020), and a broader heating anomaly leads to a smaller increase in tropical $w$. These factors, coupled with feedbacks from changes in the extra-tropics makes attribution of drivers over the long timescales of  geoengineering more challenging. However, our study suggests that the  commonly seen feature in which SRM  applied at the equator leads to an extended or permanent e-QBO state  (that is, in a state where the QBO phase is easterly in the mid-stratosphere and westerly in the lower stratosphere) is due to an increase in $w$, combined with the

QBO secondary circulation which results in a slower descent of the easterly shear zone compared to the westerly shear zone."

L. 622 feedbacks "among" maybe "between" is better

*For readability, we are happy to use 'between'.*

"The UM-UKCA simulations, which include feedbacks  between sulfate aerosol distribution, atmospheric dynamics and heating also show changes in the latitudinal structure of the QBO."

L. 644 "aerosols are"

*Thanks, this has been corrected.*

L. 672 (paragraph) In parts, this paragraph feels rather weak and just thrown out there to give a broader scope to the paper. I don't have much doubt that this research is useful, mind you, I just feel like this could be better justified. "species such as ozone and stratospheric water vapour have different signals depending on the zonal wind shear and their distribution is important for climate change projections" sounds rather weak – if there was a volcanic eruption capable of modifying the QBO, the main effect on climate change projections wouldn't be the distribution of ozone. We should know it, but the aerosols' distribution (which is affected by the QBO more than WV) would affect the climate more. Similarly, ozone is influenced by many factors in case of a volcanic eruption (see Aquila et al., 2013). "despite an increased probability of a QBO disruption in a warmer future climate" I can see why you would say this and can think of a few references for it (like Aubrey et al., 2021) but you should 1) cite them and 2) contextualize this a bit better. Seems like a shame to conclude this paper this way, the authors should work on this a bit more.

*We thank the reviewer for pointing this out and have rephrased the paragraph as follows:*

"It is now accepted that the QBO has a significant effect on the extratropical tropospheric circulation (Kidston et al., 2015; Gray et al., 2018) and this is now being taken into account in seasonal weather forecasting, exploiting the fact that the time evolution of the QBO is relatively simple and hence that the state of the QBO can be well predicted several months in advance (Scaife et al., 2014). Dynamical disruptions to the QBO, such as those observed in 2015 and 2020, significantly reduce this predictability (Newman et al., 2016; Osprey et al., 2016; Hitchcock et al., 2018), and may become more common in a future climate (Anstey et al., 2021). Similarly, disruptions to the QBO due to volcanic eruptions would reduce predictability. Furthermore, in a warmer future climate, the forcing from large (e.g. 1991 Mt. Pinatubo-magnitude) eruptions is predicted to increase due to acceleration of the BDC, which causes decreased aerosol size and increased height of eruption (Aubry et al., 2021). A larger forcing may lead to a more substantial QBO disruption, so an understanding of the impacts from such disruptions are necessary to increase societal resilience to climate change and volcanic eruptions."

Data availability – This won't do, you need to make it public by ACP guidelines.

*Thanks for your patience while we archived this data.*

*The data is available from: https://dx.doi.org/10.5285/5f7206e5854246a9ae7498305d620590.*

*Brown, F.; Marshall, L.; Haynes, P.; Schmidt, A. (2023): UM-UKCA model data for study investigating the QBO response to large tropical eruptions. NERC EDS Centre for Environmental Data Analysis, 06 April 2023. doi:10.5285/5f7206e5854246a9ae7498305d620590.*

190

**REVIEW 2:**

130: "with explosive eruptions inacted": Do you mean "enacted", or "activated"? Not sure what this is trying to say.

*This typo has been corrected to 'Enacted'.*

324: "between 40 hPa and 5 hPa" → "between 40 hPa and 15 hPa" ?

*Thank you for catching this typo. We have corrected it.*

462-463: Fig. 3e also shows a slight upward movement of the westerly QBO phase for the 60 Tg eruption (roughly months 5-8), though it's modest compared to the upward shift for the e-QBO

200    cases. The corresponding January case (Fig. S1e) is more pronounced.

*We thank the reviewer for their comments and agree that identifying an upwards movement of the zonal winds as a feature unique to the eQBO state may be incorrect so we remove this comment at L466:*

"In UM-UKCA, the $w$ anomaly forced by the aerosol heating from both 60 Tg and 15 Tg eruptions results in a complete halt of the descent of the QBO when the model is initialized in the e-QBO state but not when it is initialized in the w-QBO state (since the QBO secondary circulation for the easterly shear zone augments the aerosol effect on the upwelling, but for the westerly shear zone diminishes it). "

210   *As for the corresponding January case, we agree with the statement and this is described in the supplementary text S2 already. We add the following sentence at L319:*

'The ascent of the zonal winds is easily identified following an eruption initiated during an e-QBO, although a slight ascent may be occurring in the w-QBO case as well (Fig. 3f months 5–9). This is more pronounced for the corresponding 60 Tg eruption initiated in January (Fig. S1e).'

499: I'm not sure from Fig. 10c that the QBO westerly phase has become broader. (The broadening is also mentioned a couple times in the Conclusions.) It looks to me like it's roughly the same width but has shifted off the equator, into the NH. But I find it hard to tell just by comparing the filled and line contours in this plot. Maybe a latitudinal profile of the wind (at 20 hPa, say) would make this

220   clearer? Perhaps also with a metric of the width, e.g. full width at half maximum. However it seems to me that this result is not essential to the paper, and if the statement about broadening is not supported it could simply be removed.

*We thank the reviewer for their feedback and suggestions here. We agree that the broadening statement is unclear and is not essential to the conclusion of the paper. However, we would like to*

*make clear that the westerly winds extend farther into the SH at 10-20 hPa after a volcanic eruption (now Fig. 10e). This is attributed to a second 'peak' of westerly winds in the SH subtropics, although the main 'peak' representing the westerly phase of the QBO does not change width. To make this clear we have removed references to broadening and added some text to the supplementary.*

*Text at L511:*

230 'The QBO westerly phase has  developed a hemispheric asymmetry with its peak winds moved off the equator into the Northern Hemisphere (centered near 10° N) whereas the control westerly peaks much closer to the equator (Fig. 10e). Additionally, the zonal winds are westerly in the Southern Hemisphere between 10 and 20 hPa at 20° S after a 60 Tg eruption (Fig. 10e), but easterly in the control (Fig. 10e, dashed line contours). This structural change in the westerly wind for eruptions initiated during a w-QBO state is discussed in Supplementary Text S4.'

*Additional supplementary text:*

**"Supplementary Text 4: The latitudinal structure of the QBO**

In addition to a shift in the latitude of the QBO westerly phase, the zonal winds may also change latitudinal structure following eruptions initiated during a w-QBO state. Fig. S11a and Fig. S11b show

240 that at 20 hPa, the zonal wind of the control simulation is approximately Gaussian whereas the zonal winds after an eruption show a secondary maximum at 18° S and 5° N, respectively.

As described in Sect. 3.3, an increase in westerly wind strength at subtropical latitudes relative to the control may be related to the temperature gradient created by the aerosol distribution. Fig. S11 shows the relationship between the sAOD, change in temperature curvature and change in zonal wind compared to the control at 20 hPa for eruptions initiated during a w-QBO in July and January. The movement of the westerly phase into the Northern Hemisphere for an eruption in July (Fig. S11a) and into the Southern Hemisphere for an eruption in January (Fig. S11b) can be identified, as well as a smaller 'secondary maximum' in the zonal wind in the opposite hemisphere. The increase in zonal wind in both hemispheres coincides with a decrease in sAOD; the changes being larger in the

250 Northern Hemisphere for a July eruption and the Southern Hemisphere for a January eruption (Fig. S11c, S11d). It is likely that the decline in sulfate aerosol burden away from the equator has led to a temperature gradient that supports a westerly wind at higher latitudes. This is suggested by the increase in temperature curvature at 20 hPa, which coincides with the changes in sAOD (Fig. S11e, S11f). It is also likely that feedbacks are occurring so that changes to the zonal winds influence circulation of the aerosol, thereby sustaining the described relationship between aerosol, zonal wind and temperature.

[Figure]

**Figure S11: For a 60 Tg eruption initiated during a w-QBO, 5 months after the eruption (a), (b) show the zonal wind for the control simulation (orange dashed line) and after an eruption (orange solid line), (c), (d) show a comparison between zonal wind anomaly compared to the control simulation at 20 hPa (orange points) and sAOD (grey solid line), (e), (f) show a comparison between SAOD (grey solid line) and latitudinal temperature gradient (red solid line). Column 1 shows results for an eruption initiated in July. Column 2 shows results for an eruption initiated in January.**

260

514: "confining it more to the easterly phase": not sure what "it" refers to here.

*We have clarified to mean the sulfate aerosol.*

551-552: Because the wave forcing is weaker? (Consistent with a longer QBO period.) Perhaps worth saying this explicitly, if that's what is meant.

*We have modified the sentence as you suggest.*

270 "This may be due to differences between our simulation and the specific conditions of the observed eruptions or, perhaps more likely as we suggest below, that the QBO in UM-UKCA is more sensitive to an eruption than the observed QBO due to a weaker wave forcing, suggested by its longer period."

643: "sustaining the meridional aerosol" → "sustaining the meridional aerosol distribution" ?

*Thanks, we have corrected this sentence.*

658: "secondary circulation causes easterly shear zones to descend faster than westerly shear zones": switch "easterly" and "westerly"

*Thanks for spotting this, we have corrected the sentence.*